# Polygenic risk alters the penetrance of monogenic kidney disease

**Atlas Khan** [1], **Ning Shang** [1], **Jordan G. Nestor** [1], **Chunhua Weng** [2], **George Hripcsak**[2], **Peter C. Harris**[3], **Ali G. Gharavi** [1] & **Krzysztof Kiryluk** [1] ✉

Chronic kidney disease (CKD) is determined by an interplay of monogenic, polygenic, and environmental risks. Autosomal dominant polycystic kidney disease (ADPKD) and COL4A-associated nephropathy (COL4A-AN) represent the most common forms of monogenic kidney diseases. These disorders have incomplete penetrance and variable expressivity, and we hypothesize that polygenic factors explain some of this variability. By combining SNP array, exome/genome sequence, and electronic health record data from the UK Biobank and All-of-Us cohorts, we demonstrate that the genome-wide poly-genic score (GPS) significantly predicts CKD among ADPKD monogenic variant carriers. Compared to the middle tertile of the GPS for noncarriers, ADPKD variant carriers in the top tertile have a 54-fold increased risk of CKD, while ADPKD variant carriers in the bottom tertile have only a 3-fold increased risk of CKD. Similarly, the GPS significantly predicts CKD in COL4A-AN carriers. The carriers in the top tertile of the GPS have a 2.5-fold higher risk of CKD, while the risk for carriers in the bottom tertile is not different from the average population risk. These results suggest that accounting for polygenic risk improves risk stratification in monogenic kidney disease.

Common complex traits are determined by a combination of genetic and environmental risk factors. A small subset of common human diseases is caused by rare monogenic variants with relatively large effects that cause disease by disrupting a specific disease-related pathway[1,2]. However, monogenic disease variants typically have incomplete penetrance that is often attributable to environmental, stochastic, or other inherited factors. Genome-wide polygenic scores (GPS) have emerged as a powerful approach to quantifying the contribution of polygenic effects[3-24]. Recent studies suggested that such scores could partially explain the variable penetrance of several monogenic disorders, including familial hypercholesterolemia, hereditary breast and ovarian cancer, and Lynch syndrome[25]. However, the interplay of monogenic and polygenic risk has not been previously studied in the context of kidney disease.

Chronic kidney disease (CKD) is a common condition that affects more than 10% of the population worldwide[26]. CKD represents a

genetically complex and highly heterogeneous phenotype. Monogenic disorders account for up to 9.3% of all-cause CKD[27] with autosomal dominant polycystic kidney disease (ADPKD) and Alport syndrome, Thin Basement Membrane Disease, and Hereditary Nephritis, collectively known as collagen type IV-alpha-associated nephropathies (COL4A-AN) representing the most common forms of monogenic kidney diseases. ADPKD is caused by dominant mutations in the *PKD1* gene on chromosome 16 or the *PKD2* gene on chromosome 4. The disease affects all ancestral groups with an overall prevalence of approximately 1 in 1000[28,29]. The second most common group of inherited nephropathies, COL4A-AN, are caused by mutations in *COL4A3*, *COL4A4*, or *COL4A5* genes. COL4A-AN is characterized by glomerular basement defects manifesting with hematuria and renal dysfunction. Biallelic inheritance causes Alport syndrome, a rare and more severe disease characterized by hematuria, early-onset kidney failure, and deafness. However, monoallelic carriers of pathogenic

[1]Division of Nephrology, Department of Medicine, Vagelos College of Physicians & Surgeons, Columbia University, New York, NY, USA. [2]Department of Biomedical Informatics, Vagelos College of Physicians & Surgeons, Columbia University, New York, NY, USA. [3]Division of Nephrology and Hypertension, Mayo Clinic, Rochester, MN, USA. ✉e-mail: kk473@columbia.edu

variants also have a higher risk of CKD. The penetrance of both ADPKD and COL4A-AN is highly variable, even within the same pedigrees. Polygenic background may partially explain the observed variability in the penetrance of these disorders.

In this work, we hypothesize that monogenic variants exert substantial effects by disrupting critical disease pathways, while polygenic risk factors may either mitigate or exacerbate these effects by influencing a broader range of mechanisms associated with CKD. We have previously developed a GPS for CKD with a validated performance across diverse ancestries[30]. Here, we test if this GPS determines the risk of CKD among carriers of pathogenic ADPKD and COL4A-AN variants through combined analysis of exome/genome sequence, SNP microarray, and electronic health record (EHR) data for 568,457 participants of the UK Biobank (UKBB) and the All of Us (AoU) study. We demonstrate that the GPS is significantly associated with a higher risk of renal dysfunction among ADPKD as well as COL4A-AN monogenic variant carriers.

## Results

The summary of our analytical approach is provided in Fig. 1. Using our electronic phenotyping strategy (see Methods), we defined a total of 10,081 CKD cases and 266,724 controls in the UKBB and 11,820 CKD cases and 22,763 controls in the AoU dataset. All of these participants met our strict inclusion/exclusion criteria and had both high-quality sequence and SNP genotype data available for analysis.

### Autosomal dominant polycystic kidney disease (ADPKD)

We first identified all *PKD1* and *PKD2* variants that were either pLoF or reported as pathogenic ('P') by at least two ClinVar submitters without conflicts (model M1). A total of 172 and 34 carriers of such variants were found in the UKBB and AoU, corresponding to the overall prevalence of approximately 0.036% and 0.034%, respectively. We performed a Meta-PheWAS analysis of both UKBB and AoU datasets to assess phenome-wide associations of M1 variants (Fig. 2a). The top associated phecode was "Cystic Kidney Disease" with OR = 295.7 (95% CI: 214.3–408.0, $P = 9.0E-263$), as expected. We also detected

significant associations with a variety of CKD-related phecodes, including "End-stage renal disease", OR = 52.8 (95%CI: 31.2–89.3, $P = 2.1E-49$) and "Kidney replaced by transplant" OR = 112.1 (95%CI: 71.5–175.7, $P = 4.9E-94$), as well as multiple other renal and extra-renal complications of ADPKD (Supplementary Data 3), confirming that M1 variant definitions have robust phenotypic signatures across both biobanks. Additional sensitivity analyses demonstrated that these results were not biased by ancestry and were consistent for different variant models and individual genes (Supplementary Fig. 6). We next tested the effects of M1 variants on the risk of CKD, as defined by our phenotyping algorithm, after adjustment for age, sex, diabetes, batch, and ancestry (Supplementary Table 6). In the meta-analysis of both cohorts, the risk of CKD was 17-fold higher in the ADPKD M1 variant carriers compared to noncarriers (OR: 17.1, 95%CI: 11.1–26.4, $P = 1.8E-37$).

We next investigated the effect of polygenic background on the risk of CKD by computing our previously validated GPS for CKD[30] in all UKBB and AoU participants. After *APOL1* and ancestry adjustments, the polygenic score was standard normal-distributed across ancestries in both UKBB and AoU datasets (Supplementary Fig. 2). Because this risk score has not been previously tested in AoU participants, we first confirmed that the GPS was indeed associated with increased risk of CKD in this dataset (OR per SD = 1.39, 95%CI: 1.36–1.43, $P = 5.9E-125$, adjusted for age, sex, diabetes, batch, and genetic ancestry). All participants were then stratified based on their ADPKD QV carrier status, and the effects of the GPS were re-examined within each stratum across both UKBB and AoU datasets combined. In the meta-analysis, the OR per SD of the GPS was 2.28 (95%CI: 1.55–3.37, $P = 2.7E-05$) in the M1 QV carriers and 1.72 (95%CI: 1.69–1.76, $P < E-300$) in the noncarriers (Table 1). Despite the trend for a greater effect of the GPS among the carriers, the GPS-by-carrier interaction test was not statistically significant in either cohort or in the combined meta-analysis (Supplementary Table 7).

We next estimated the CKD risk for each tertile of the GPS distribution among the M1 variant carriers compared to the middle tertile of the noncarriers (i.e., reflecting average population risk) across both

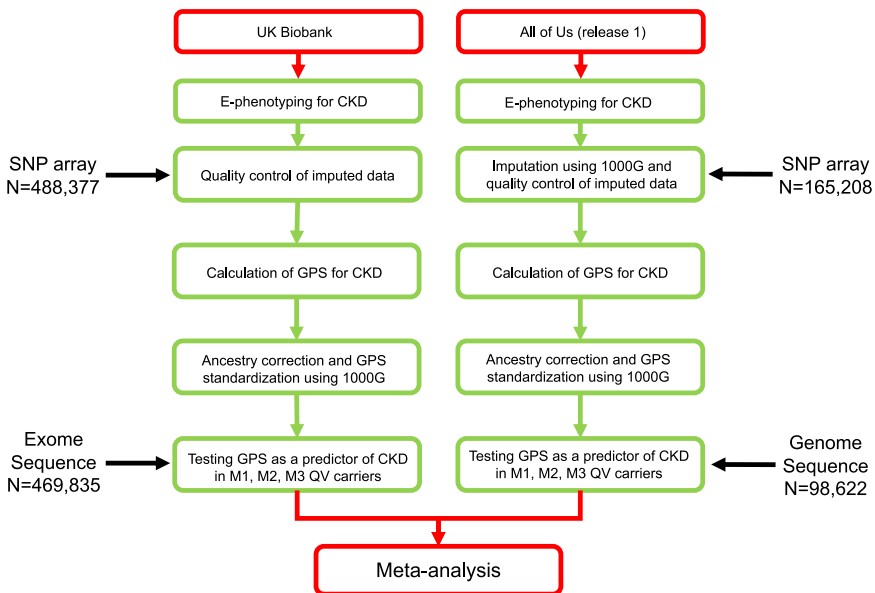

**Fig. 1 | Overview of the workflow for the analysis of phenotype and genotype data.** The analysis involved genotype and phenotype data from the UK Biobank (left) and All of Us Study (right). Electronic phenotyping for chronic kidney disease (CKD) was performed in both datasets. The All-of-Us genotype data were additionally imputed using 1000 Genomes reference. In both datasets, genome-wide polygenic scores (GPS) for CKD were calculated using the same method, corrected for ancestry, and standardized using 1000 Genomes reference. The M1, M2, and M3 variants were defined using the same definitions based on exome sequence data in the UK Biobank and genome sequence in the All of Us dataset. The joint analyses of GPS and M1, M2, M3 variants (see Methods) were performed in each dataset, and summary statistics were meta-analyzed using the fixed-effects model across both biobanks.

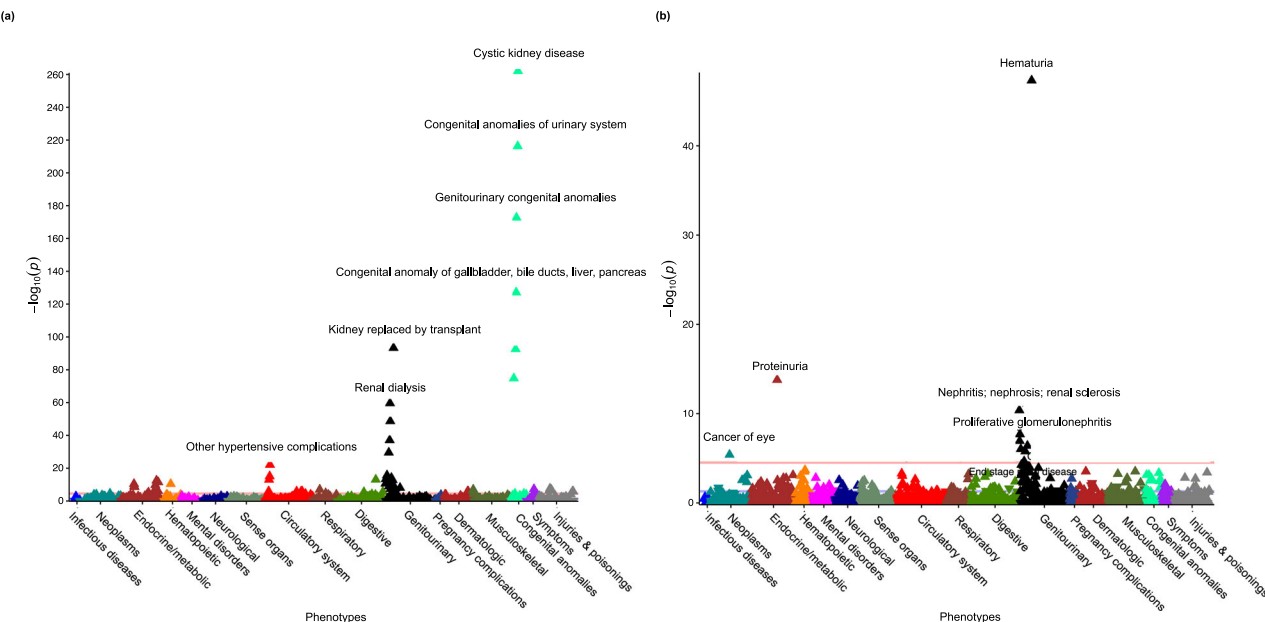

**Fig. 2 | Phenome-wide meta-analysis (Meta-PheWAS) for ADPKD and COL4A-AN M1 carriers.** Meta-PheWAS for **a** ADPKD M1 variant carriers and **b** COL4A-AN M1 variant carriers. The analysis includes combined data from 460,360 UKBB and 74,350 AoU participants, with both genotype and phenotype data available. Both analyses were conducted under a dominant inheritance using logistic regression adjusted for age, sex, batch, and ancestry. The effect estimates and two-sided P- values were generated using fixed-effects meta-analysis of individual cohorts. The red horizontal lines indicate a phenome-wide significance level after accounting for the number of phecodes tested (*P* = 2.8E-05). Y-axis: -log10(*P*-value) from fixed-effects meta-analysis (two-sided and not adjusted for multiple testing). X-axis: system-based phecode groupings. An upward-pointing triangle indicates increased odds for a given phecode, and a downward-pointing triangle indicates reduced risk.

AoU and UKBB (Fig. 3 and Supplementary Table 8). Notably, ~34% of QV carriers had CKD stage 3 or above compared to ~3% of noncarriers in the middle GPS quartile. Among the QV carriers, we observed a clear gradient of CKD risk as a function of GPS, ranging from OR = 3.03 (95% CI 1.03–8.95, *P* = 4.4E-02) for the lowest tertile to OR = 54.4 (95%CI 26.1–113.0, *P* = 9.6E-27) for the highest tertile of polygenic risk. These results demonstrate that the GPS partially accounts for the incomplete penetrance of M1 qualifying variants.

**Sensitivity analyses**

In the subgroup analyses, we examined QVs in *PKD1* and *PKD2* separately and observed similar patterns of GPS effects within each of the gene-defined subgroups (Supplementary Fig. 3). Similarly, we examined QVs by variant type (truncating vs. missense) and observed a consistent pattern of GPS effects for both subgroups (Supplementary Fig. 4). Lastly, we investigated the effect of the GPS on the risk of CKD among ADPKD carriers defined under two alternative QV models (M2 and M3, Supplementary Table 8). Similar results on the penetrance of CKD were observed, demonstrating that our findings were also robust to less stringent QV definitions. We also tested for the effect of the new race-free CKD-EPI eGFR formula[31] to define cases and controls but observed similar results despite the smaller number of cases (Supplementary Table 9a). We repeated this analysis, including only UKBB participants of European ancestry, and the trends remained significant regardless of the eGFR equation used (Supplementary Table 10a).

**Collagen IV alpha-associated nephropathy (COL4A-AN)**

We next examined the effect of GPS on the risk of CKD in the carriers of COL4A-AN variants compared to the average risk of noncarriers. In this analysis, we used a less stringent MAF < 0.001 for variant filtering, considering that the most severe phenotype of COL4A-AN is observed under a recessive model. Under M1, we defined a total of 1435 carriers in the UKBB and 310 carriers in the AoU dataset, corresponding to the overall prevalence of approximately 0.31% and 0.32%, respectively.

In the Meta-PheWAS analysis for M1 carriers across both UKBB and AoU datasets (Fig. 2b), the top associated phecode was "Hematuria" with OR = 2.3 (95% CI: 2.0–9.6, *P* = 4.8E-48). Other phenome-wide-significant associations included "Kidney replaced by transplant" (OR = 3.1, 95%CI: 2.0–23.8, *P* = 3.8E-07), "Nephritis, nephrosis, renal sclerosis" (OR = 2.34, 95%CI: 1.81–10.39, *P* = 4.1E-11), "Proteinuria" (OR = 3.94, 95%CI: 2.77–51.6, *P* = 1.6E-14) and "Chronic glomerulonephritis, NOS" (OR = 2.98, 95%CI: 1.92–19.7, *P* = 9.0E-07). The complete list of phenotypic associations is provided in Supplementary Data 4. Sensitivity analyses demonstrated that these results were not biased by ancestry and were consistent for different variant models and individual genes (Supplementary Fig. 7). Compared to noncarriers, the M1 QV carriers had a 37% increased risk of CKD as defined by our e-phenotype (OR = 1.37, 95%CI: 1.13–1.64, *P* = 8.5E-04), M2 carriers had 25% increased risk (OR = 1.25, 95%CI: 1.00–1.56, *P* = 4.9E-02), and M3 carriers had 48% increased risk (OR = 1.48, 95%CI: 1.23–1.77, *P* = 2.6E-05) in the combined meta-analysis under a dominant model (Supplementary Table 11). In comparison, the M3 recessive genotype was associated with a 3.38-fold higher risk (OR = 3.38, 95%CI: 1.88–6.08, *P* = 4.7E-05).

We next investigated the effect of polygenic background on the risk of CKD among M1 QV carriers compared to noncarriers. Similar to ADPKD, the GPS had a significant effect on the risk of CKD among both COL4A-AN carriers (OR per SD of GPS = 1.78, 95%CI: 1.22–2.58, *P* = 2.4E-03) and noncarriers (OR per SD of GPS = 1.70, 95%CI: 1.68–1.73, *P* < E-300) in the meta-analysis (Table 1). There was no significant GPS-by-carrier interaction (*P* = 8.1E-01) (Supplementary Table 12). Approximately 8% of M1 variant carriers had CKD stage 3 or above compared to only 3% of noncarriers in the middle GPS quartile. Similar to ADPKD, we observed a gradient of CKD risk as a function of the GPS among M1 carriers, from no increased risk (OR = 1.01, 95%CI 0.63–1.86, *P* = 7.8E-01) for the lowest GPS tertile to a 2.5-fold higher risk (OR = 2.53, 95%CI 1.66–3.85, *P* = 1.4E-05) for the top GPS tertile when compared to the middle tertile of noncarriers (Fig. 4).

**Table 1 | Performance metrics for the genome-wide polygenic score (GPS) in ADPKD and COL4A-AN M1, M2, and M3 carriers and noncarriers**

| Model | Cohort | Cases/controls | CKD GPS OR per SD (95% CI), P | AUC full model (95%CI) | AUC crude (95%CI) | Variance explained |
|---|---|---|---|---|---|---|
| ADPKD | | | | | | |
| Noncarrier | | | | | | |
| | UKBB | 9565/252,870 | 1.80 (1.76–1.84), P < E-300 | | | |
| | AoU | 11,830/22,773 | 1.40 (1.36–1.44), P = 8.5E-211 | | | |
| | Meta | 21,395/275,643 | 1.72 (1.69–1.76), P < E-300 | 0.78 (0.78–0.78) | 0.62 (0.62–0.62) | 0.039 |
| M1 Carrier | | | | | | |
| | UKBB | 36/79 | 2.45 (1.37–4.38), P = 2.6E-03 | | | |
| | AoU | 5/2 | 3.49 (0.22–54.30), P = 3.7E-01 | | | |
| | Meta | 41/81 | 2.28 (1.55–3.37), P = 2.6E-05 | 0.96 (0.92–1.00) | 0.69 (0.59–0.79) | 0.128 |
| M2 Carrier | | | | | | |
| | UKBB | 39/86 | 2.19 (1.31–3.64), P = 2.6E-03 | | | |
| | AoU | 5/9 | 2.47 (0.58–10.50), P = 2.2E-01 | | | |
| | Meta | 44/95 | 2.21 (1.37–3.58), P = 3.3E-05 | 0.97 (0.93–1.00) | 0.70 (0.60–0.80) | 0.103 |
| M3 Carrier | | | | | | |
| | UKBB | 45/211 | 5.10 (2.10–12.40), P = 3.2E-04 | | | |
| | AoU | 7/4 | 1.18 (0.34–4.13), P = 7.9E-01 | | | |
| | Meta | 52/215 | 5.25 (2.31–11.9), P = 7.4E-05 | 0.97 (0.94–1.00) | 0.69 (0.60–0.78) | 0.076 |
| COL4A-AN | | | | | | |
| M1 Carrier | | | | | | |
| | UKBB | 62/1152 | 1.93 (1.26–2.95), P = 2.3E-03 | | | |
| | AoU | 37/41 | 1.35 (0.62–2.94), P = 4.5E-01 | | | |
| | Meta | 99/1193 | 1.78 (1.22–2.58), P = 2.4E-03 | 0.94 (0.91–0.97) | 0.59 (0.52–0.65) | 0.019 |
| M2 Carrier | | | | | | |
| | UKBB | 65/1285 | 2.37 (1.48–3.80), P = 3.2E-04 | | | |
| | AoU | 47/59 | 3.09 (1.13–8.46), P = 2.7E-02 | | | |
| | Meta | 112/1344 | 2.47 (1.56–3.94), P = 1.3E-04 | 0.93 (0.90–0.96) | 0.62 (0.56–0.68) | 0.014 |
| M3 Carrier | | | | | | |
| | UKBB | 100/1730 | 1.66 (1.29–2.13), P = 8.9E-05 | | | |
| | AoU | 72/154 | 1.48 (1.13–1.94), P = 4.3E-03 | | | |
| | Meta | 172/1884 | 1.57 (1.31–1.89), P = 1.5E-06 | 0.89 (0.86–0.92) | 0.60 (0.55–0.65) | 0.019 |

Odds ratios (OR) per standard deviation (SD) of the GPS were estimated using logistic regression adjusted for age, sex, diabetes, batch, and genetic ancestry. Meta OR were estimated using fixed-effects meta-analysis of adjusted effect estimates derived from individual cohorts. For individual cohorts, the P-values correspond to the Wald test from logistic regression, and Meta P-values correspond to fixed-effects meta-analyses. All P-values are two-sided and not corrected for multiple testing. AUC was calculated for the full model (GPS and covariates) and for GPS alone without covariates (crude); variance explained was calculated for the GPS alone by estimating variance explained by the full model (GPS and covariates) minus the variance explained by the covariates-only model.
CI confidence intervals.

We also explored the recessive model by testing for GPS effects among individuals with the M3 risk genotype (QV homozygotes, compound heterozygotes, or *COL4A5* hemizygous males). For individuals with the risk genotype, the top tertile of the GPS conveyed a 6.73-fold higher risk of CKD (OR = 6.73, 95%CI: 2.59–17.5, P = 8.8E-05), while the bottom tertile conveyed a 2.29-fold higher risk of CKD (OR = 2.29, 95%CI 0.64–8.12, P = 2.0E-01) compared to the middle tertile of individuals without the risk genotype (Supplementary Fig. 5).

**Sensitivity analyses**
Our sensitivity analyses included alternative variant models (Supplementary Table 13) and separate analyses of autosomal (*COL4A3* and *COL4A4*) and sex-linked (*COL4A5*) genes (Supplementary Table 14). These analyses confirmed the direction-consistent effect of the GPS across all different subgroups. We note that recessive analyses for M1 and M2 models were underpowered due to the low overall frequency of recessive genotypes defined under these models. We also tested for the effect of the new CKD-EPI eGFR equation[31] but observed no changes in the GPS performance (Supplementary Table 9b), and our results were consistent when the analysis was limited to the UKBB participants of European ancestry (Supplementary Table 10b).

## Discussion
Our large-scale analyses of UKBB and AoU datasets demonstrated that polygenic background has an effect on the risk of kidney disease among individuals with the most common forms of monogenic kidney disorders. Among the individuals with known pathogenic or rare pLOF variants in *PKD1* or *PKD2*, the bottom tertile of the GPS was associated with a 3-fold increased risk compared to the middle tertile of noncarriers (average risk). In contrast, the top tertile was associated with a 54-fold increased risk of CKD compared to the average risk. Similar but less extreme patterns were also observed for COL4A-AN. The carriers of known pathogenic or rare pLOF variants in COL4A-AN genes in the bottom tertile of the GPS had no increased risk of CKD, while the individuals in the top GPS tertile had a 3-fold higher risk of CKD compared to noncarriers. Under the recessive model, the risk was 2-fold higher and nearly 6-fold higher for the bottom and top tertile of the GPS, respectively, compared to the average risk of noncarriers.

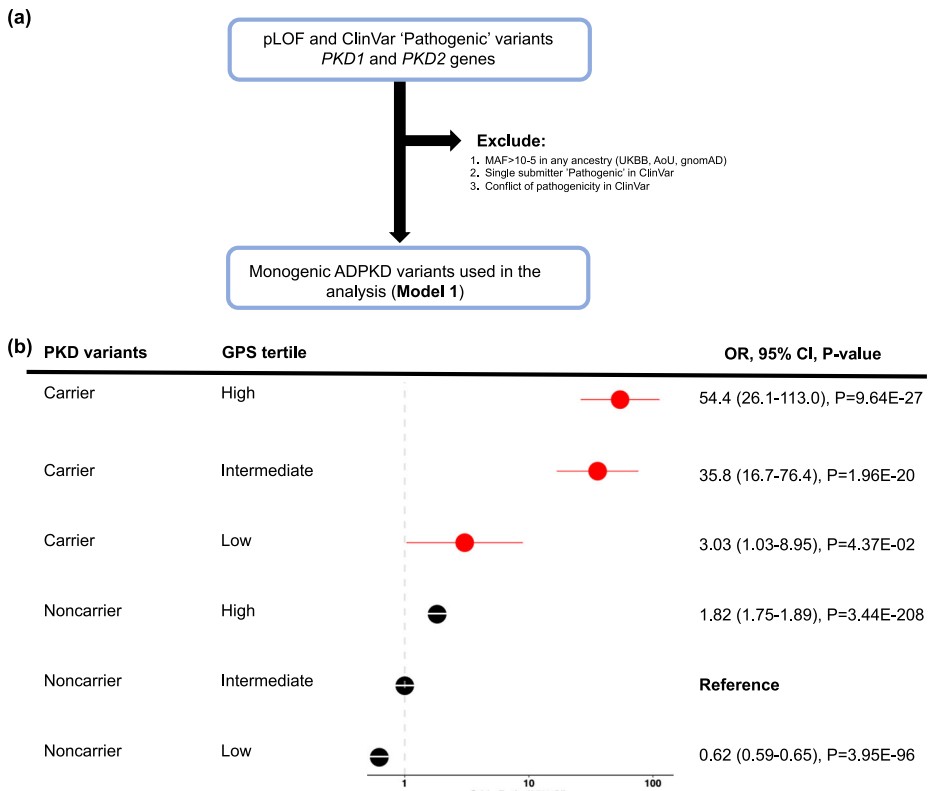

**Fig. 3 | Polygenic effects on the risk of CKD among ADPKD M1 variant carriers (dominant model). a** M1 qualifying variant filtering strategy; **b** CKD risk for each polygenic risk score tertile compared to the middle tertile of noncarriers (average population risk). The analysis includes $N = 262,435$ UKBB participants ($N_{cases} = 9565$ and $N_{controls} = 252,870$) and $N = 34,603$ AoU participants ($N_{cases} = 11,830$ and $N_{controls} = 22,773$). The noncarriers with intermediate polygenic scores (middle tertile) served as the reference group for all calculations. The $X$-axis shows odds ratios (OR); the dotted vertical line corresponds to the OR = 1.0 (no change in risk compared to the reference). The odds ratios were estimated by the fixed-effects meta-analysis of individual cohort (UKBB and AoU) estimates obtained using logistic regression with adjustment for age, sex, batch, and ancestry. The circles correspond to adjusted OR, and horizontal lines correspond to 95% confidence intervals (CI). Two-sided $P$-values were derived using fixed-effects meta-analysis and are not corrected for multiple testing. GPS genome-wide polygenic score.

While our analyses suggest that the GPS alters the penetrance of renal dysfunction in ADPKD and COL4A-AN, we recognize that our study has limitations. First, significant demographic differences exist between the UKBB and the AoU participants. The UKBB participants are older (mean age 56.5 years, range 40–69 years) and predominantly (94%) of European ancestry, while the AoU participants are younger (mean age 54.9 years, range 18–89 years) and have more diverse ancestral backgrounds (57% non-European). Although our GPS has improved cross-ancestry portability, the performance is still lower in individuals of African compared to European ancestry. This may explain the observation of lower GPS effects in the AoU dataset compared to the UKBB, but the demographic differences and other unmeasured exposures may also be contributing. Moreover, current catalogs of "P" and "LP" variants are more comprehensive for European compared to non-European genomes. Thus, we are also more likely to misclassify pathogenic variants in the AoU dataset compared to the UKBB dataset, and such misclassification could have reduced the observed effect sizes.

Second, we were able to investigate only the two most common forms of monogenic kidney diseases, ADPKD and COL4A-AN. Similar patterns of GPS effects observed in these very different disorders suggest that our findings may be generalizable to other less frequent monogenic kidney diseases. However, much larger datasets would be needed to validate this hypothesis. Moreover, we are underpowered in our tests for interactions between monogenic and polygenic risks, and this issue would also be addressed by larger datasets.

Third, we are aggregating qualifying variants across all known genes for ADPKD or COL4A-AN. However, the penetrance of kidney disease is known to vary according to a specific gene (e.g., *PKD2* vs. *PKD1*) or a specific mutation type (e.g., missense vs. truncating variants). We performed sensitivity analyses to address this issue, and our analyses by gene and variant type demonstrated consistent patterns of GPS effects across all subgroups. At the same time, we note that some of our subgroup analyses were underpowered. For example, *PKD2* mutations account for only ~15–20% of ADPKD cases and lead to a less severe disease compared to *PKD1*[32], impacting our power for individual analysis of this gene. Similarly, we do not have adequate power to define GPS effects under recessive inheritance using our most stringent (M1 and M2) models in COL4A-AN. Thus, our biallelic analysis was performed only for the M3 model.

Fourth, there are notable limitations regarding kidney disease phenotyping in large biobanks related to ascertainment biases, the cross-sectional nature of data, the non-random missingness of EHR diagnoses, and the inability to perform manual chart reviews to confirm the diagnosis. These and other limitations of our e-phenotyping strategy have been discussed elsewhere[33]. We utilized the 2009 CKD-EPI equation in our primary analysis for consistency with our earlier work[30,34]. However, our sensitivity analysis confirmed the results when cases and controls were re-defined using the new 2021 CKD-EPI equation[31], demonstrating that our conclusions are robust to the equation choice. Moreover, complete albuminuria and hematuria data were not uniformly available for all individuals included in our analyses, precluding GPS testing against these disease manifestations. This is particularly relevant to COL4A-AN, which most commonly presents only with hematuria and proteinuria, and could partially explain why the GPS effect was less pronounced in COL4A-AN compared to ADPKD.

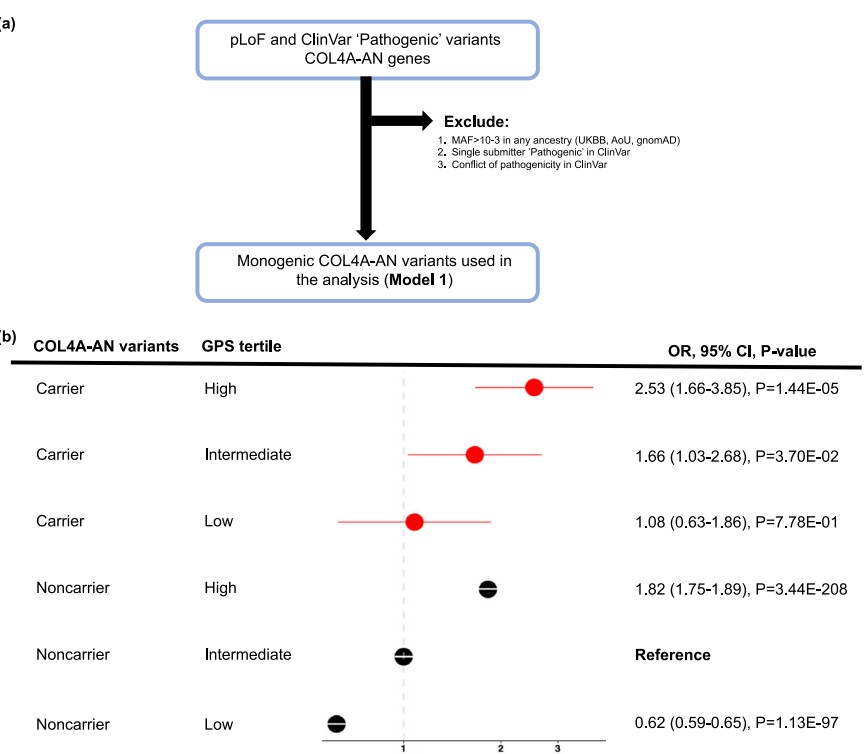

**Fig. 4 | Polygenic effects on the risk of CKD among M1 carriers of COL4A-AN variants (dominant model). a** M1 qualifying variant filtering strategy; **b** CKD risk for each polygenic score tertile compared to the middle tertile in noncarriers (average population risk). The analysis includes $N = 262,435$ UKBB participants ($N_{cases} = 9565$ and $N_{controls} = 252,870$) and $N = 34,603$ AoU participants ($N_{cases} = 11,830$ and $N_{controls} = 22,773$). The noncarriers with intermediate polygenic risk (middle tertile) served as the reference group for all calculations. The $X$-axis shows odds ratios (OR); the dotted vertical line corresponds to the OR = 1.0 (no change in risk compared to the reference). The odds ratios were estimated by the fixed-effects meta-analysis of individual cohort (UKBB and AoU) estimates obtained using logistic regression with adjustment for age, sex, batch, and ancestry. The circles correspond to adjusted OR, and horizontal lines correspond to 95% confidence intervals (CI). Two-sided $P$-values were derived using fixed-effects meta-analysis and are not corrected for multiple testing. GPS genome-wide polygenic score.

The notable strength of our phenotyping approach, however, is the fact that we are able to combine structured billing code data with the available laboratory tests to not only define CKD cases but also to uniformly stage the degree of renal dysfunction with a high degree of confidence.

Lastly, we recognize several important limitations of the GPS for CKD that was used here as a proxy for polygenic effects. Even though the performance of our GPS has been previously optimized for cross-ancestry prediction, the portability could be further improved using larger and more diverse GWAS for renal function and newer statistical methods[35,36]. These and other limitations of our GPS have previously been discussed in depth elsewhere[30]. The effects of monogenic kidney disease demonstrated here will need to be re-assessed once more powerful polygenic scores for CKD become available.

In summary, in our combined analysis of exome/genome sequencing, SNP microarray, and EHR data, we observed significant independent and additive effects of monogenic and polygenic factors on the risk of kidney disease across two large-scale biobanks. We conclude that polygenic risk scores could potentially improve current clinical risk stratification in ADPKD and COL4A-AN. Testing the generalizability of these findings to other forms of inherited kidney disorders will require further studies.

## Methods
### Ethics statement
This research study involves the analysis of fully de-identified data and complies with all relevant ethical regulations as reviewed and approved by the Columbia University Institutional Review Board (Protocol # IRB-AAAC7385).

### Study design
This cross-sectional study involves a combined analysis of the UKBB and AoU cohorts. All participants provided informed consent to participate in genetic studies. Each cohort was first analyzed separately, and cohort-specific results were combined using fixed-effects meta-analysis.

### UK Biobank (UKBB)
The UKBB is a longitudinal cohort of individuals ages 40–69 years at enrollment, recruited between 2006 and 2010 across the United Kingdom[37]. The individuals recruited to UKBB signed an electronic consent to allow the broad sharing of their anonymized data for health-related research. UKBB generated and released SNP microarray, exome sequence, and structured EHR data for 469,835 participants. The cohort is 54% female, with a mean age of 57 years, and the composition is 94% Europeans, 2% West or Southeast Asians, and 2% African ancestry by self-report[37] (Supplementary Table 1).

**SNP microarray data.** The details of the UKBB microarray genotyping, imputation, and quality control are available elsewhere[37]. Briefly, using the UKBB Axiom Array ($N = 438,427$) and UK BiLEVE Axiom Array ($N = 49,950$), a total of 488,377 participants have been genotyped for 805,426 overlapping markers. The 1000 Genomes, UK10K, and Haplotype Reference Consortium (HRC) reference panels were used to perform genome-wide imputation using IMPUTE2 software[38,39]. We performed post-imputation quality control analyses as described in our previous work based on this dataset[30] retaining 9,233,643 common (i.e., Minor Allele Frequency (MAF) > 0.01), high-quality (imputation $R^2 > 0.80$) variants for the purpose of GPS calculation. To eliminate any

potential confounding by close familial relationships, we excluded cryptically related individuals (kinship coefficient > 0.0442)[40] from downstream analyses.

**Exome sequencing.** The exome sequencing (ES) dataset was generated for $N = 469,835$ UKBB participants as previously described[41,42]. Briefly, ES was performed at the Regeneron Genetics Center using 75 base pair (bp) paired-end reads with 10 bp index reads on the Illumina NovaSeq 6000; the reads were mapped to the Genome Reference Consortium Human ref. 38 (GRCh38) using the BWA-MEM command for each sample. WeCall was used to identify variants in gVCFs, which were then aggregated with GLnexus into a joint-genotyped and multi-sample project-level VCF (pVCF). SNV and indel genotypes called threshold read depth (DP) were less than 7 and 10, respectively. Subsequent variant-level filters include at least one homozygous variant carrier or at least one heterozygous variant carrier with an allele balance greater than 0.15 for SNVs and 0.20 for indels[41,42]. We accessed and analyzed the latest data through the UKBB Research Analysis Platform (RAP) on DNAnexus. For the purpose of this study, we applied additional variant-level filters that included genotype quality (GQ) > 90, depth of coverage (DP) > 10, and MAF less than or equal to 0.00001 for ADPKD and 0.001 for COL4A-AN variants in the UKBB and GNOMAD database for each ancestry[43].

**Genetic ancestry analysis.** We used the UKBB genotype array data to perform principal component analysis (PCA). We first pruned the genotype data using the plink command '--indep-pairwise 500 50 0.05'. We then used FlashPCA[44] based on 35,091 pruned variants. We merged the UKBB samples with 2504 participants of the 1000 Genomes Project (1KG phase 3)[45] and kept only shared variants between the two datasets. Then, we used a random forest machine learning based on 10 principal components to train ancestry classifiers using 1KG labeled data. Finally, we used the trained model to predict the genetic ancestry of the UKBB samples (Supplementary Fig. 1a, b).

### All of Us (AoU)

The AoU research program launched recruitment in 2018 across 340 sites across the United States, and over 372,380 participants were enrolled by 2022. AoU combines participant-derived data from surveys such as self-reported health information, physical measurements, electronic health records, and biospecimens. We analyzed the AoU data on Workbench, a cloud-based environment[46]. The first data release included $N = 98,622$ participants with complete SNP microarray and genome sequencing data as well as phenotype information. The participants included 60% female, the mean age was 55 years and consisted of 53% European, 4% Asian, and 21% Black/African American race by self-report. In addition, 17% of the cohort self-reported Hispanic/Latinx ethnicity (Supplementary Table 1).

**SNP microarray genotype data.** All participants were genotyped with the Illumina Global Diversity Array (GDA). This microarray contains 1,904,679 SNVs and 44,172 indels. First, we performed genome-wide imputation analysis on the Workbench platform. Before imputation, we excluded all variants with MAF ≤ 0.005 (671,685 variants) or genotype missingness rate ≥ 0.05 (41,526 variants). The genomic positions were lifted over from human GRCh38 to hg19 for 96% of SNPs. We then adopted the TopMed pre-imputation quality control (QC) pipeline to correct allele designations and additionally remove poorly mapping variants[47]. After QC, we used 1,191,468 variants for imputation. To reduce RAM usage and increase speed, we split the 165,208 subjects with microarray data into 8 equal batches and then imputed each batch separately. After pre-phasing with EAGLE v.2[48], we imputed missing genotypes using Minimac4[38] and 1KG phase 3v5[45] reference panel. A total of 43,371,225 autosomal variants were imputed in 165,208 individuals (Supplementary Table 2). We then merged the

eight batches based on position using VCFtools software with the command 'vcftools --gzvcf --positions --recode --recode-INFO-all –stdout'. MAFs for the imputed markers were closely correlated (correlation coefficient (r) = 0.96) with the MAFs for the 1KG dataset.

**Genetic ancestry analysis.** Similar to the UKBB data, we first pruned the genetic data using the command '--indep-pairwise 500 50 0.05' in PLINK[49] and used $N = 36,358$ pruned variants for kinship and ancestry analysis. Using KING software[40], we removed 270 samples with pairwise kinship coefficients>0.35. We then merged our AoU samples with 1KG samples, kept only SNPs in common between the two datasets, calculated PCs for the 1KG samples, and projected each of our samples onto those PCs. We then used a random forest-based machine learning approach to assign a continental ancestry group to each AoU sample. Briefly, we trained and tested the random forest algorithm on 1KG subjects with known labels. We trained the random forest model using 10 PCs as a labeled feature matrix. Then, we used our trained random forest model to predict the genetic ancestries for the AoU dataset (Supplementary Table 3 and Supplementary Fig. 1c, d).

**Whole genome sequencing.** We utilized 98,622 whole genome sequencing (GS) data released on March 15, 2020. A detailed description of GS is available elsewhere[50]. Briefly, the GS data were generated with NovaSeq 6000. DRAGEN v3.4.12 (Illumina) was used for genome alignment and calling, providing 702,668,125 SNVs for 98,622 samples with mean coverage greater or equal to 30x and >90% of bases at 20x coverage. The GS data is available in the All of Us workbench in the Hail matrix. We extracted all variants in *PKD1*, *PKD2*, *COL4A3*, *COL4A4*, and *COL4A5* genes in VCF format using the following hail command in Jupyter Notebook:

```
Gene_intervals = ['chr16:2.10M-2.15 M', 'chr4:87M-89M','chr2:220M-235M','chrX:107M-109M']
mt = hl.filter_intervals
(mt, [hl.parse_locus_interval(x,)
for x in Gene_intervals])
hl.export_vcf(mt, output_location, tabix=True)'
```

We then converted the vcf format data to the bed/bim/fam format using PLINK software[49].

### Rare variant quality control, filtering, and classification

We analyzed genetic variants in protein-coding regions of two ADPKD genes (*PKD1* and *PKD2*) and three COL4A-AN genes (*COL4A3*, *COL4A3*, and *COL4A5*) in the UKBB and AoU datasets. We first removed variants with low genotype quality (GQ < 90), depth of coverage (DP < 10), and synonymous variants. Next, we filtered variants based on frequency, excluding variants with MAF > 0.00001 for *PKD1* and *PKD2* (considering autosomal dominant inheritance of ADPKD) and MAF > 0.001 for *COL4A3*, *COL4A4*, and *COL4A5* (considering recessive inheritance of the most severe COL4A-AN phenotypes) in any ancestral group across the UKBB, AoU, and gnomAD datasets[51]. We next used a range of prediction scores to define qualifying variants (QV), as recently proposed[41]. First, we identified all rare predicted loss of function (pLOF) variants, including stop-gain, frameshift, stop-lost, start-lost, and essential splice variants. Second, we classified rare missense variants as deleterious if they met the following strict criteria: (1) Revel score > 0.70[52] and (2) variants predicted as damaging by the consensus of five predictors: Sorting Intolerant from Tolerant (SIFT)[53], Polymorphism Phenotyping v2 (PolyPhen2) HDIV and PolyPhen2 HVAR[54]; likelihood ratio test (LRT)[55]; and MutationTaster[56]. After defining the lists of pLOFs and predicted deleterious missense variants, we intersected these variants with ClinVar and Varsome databases and excluded all variants previously reported as 'Benign' (B) or 'Likely Benign' (LB) by at least one of these databases[57,58]. Third, we identified all additional rare variants reported as 'Pathogenic' (P) or 'Likely Pathogenic' (LP) by at least two independent ClinVar submitters (accessed

on 11/13/22). To increase the specificity, we excluded any variants with a conflict of reported pathogenicity or those submitted to ClinVar by only a single submitter. Based on these annotations, we then analyzed the data defining carrier status by three distinct variant classification models: the most stringent model (M1) included only pLOF and reported 'P' variants as defined above; model 2 (M2) was relaxed also to include pLOF, 'P', and 'LP' variants; and model 3 (M3) was further relaxed to include pLOF, and all deleterious missense variants predicted as deleterious by all 5 algorithms, with revel score >0.7, and not previously classified as 'B' or 'LB' by ClinVar. We defined the penetrance of M1, M2, and M3 models as the probability of CKD conditioned on the QV carrier status. Notably, while M2 contains all M1 variants, the definition of M3 does not necessarily encompass all M1 and M2 variants. The list of observed qualifying variants included under each model is provided in Supplementary Data 1 and 2. Because the biallelic inheritance of pathogenic variants in *COL4A* genes causes Alport syndrome (the most severe form of COL4A-AN), we additionally analyzed recessive inheritance by defining homozygous or compound heterozygous (*COL4A3* and *COL4A4*) or hemizygous (for *COL4A5* in males) genotypes for the qualifying variants.

### Genome-wide polygenic score (GPS)

We used the GPS for CKD previously validated across diverse ancestries[30]. This GPS is based on the *P*-value thresholding (P+T) method and involves 41,426 common autosomal markers with nonzero weights selected based on $r^2 \leq 0.2$ and $P \leq 0.03$ from the original GWAS for renal function. The GWAS used for the development of this GPS was based on different cohorts than the ones used for optimization and testing. The score was calculated using the PLINK command '--bfile --score sum --out' based on imputed genotype data. The GPS distribution was ancestry-adjusted for mean and variance based on 1KG reference, normal standardized, and additionally adjusted for *APOL1* risk genotype as previously proposed (Supplementary Fig. 2)[30]. The *APOL1* risk alleles were imputed for all subjects, and the risk genotype was defined under a recessive model as G1G1, G2G2, or G1G2 risk allele combinations across all datasets (Supplementary Table 4). Because the original GPS model was selected based on the optimization step involving 70% of UKBB participants and including ADPKD/ COL4A-AN QV carriers, in our sensitivity analyses, we excluded all QV carriers from this dataset ($N = 1373$) and re-optimized the score using exactly the same methods and models as in our original publication[30]. We note that for the P+T method, the optimization step affects only SNP selection and not SNP weights since the weights are based on the original discovery GWAS and remain fixed for all P+T models. This analysis confirmed that the same GPS model as originally proposed had superior performance over all other models and, in fact, showed slightly better performance when compared to the original analysis (Supplementary Table 5). These analyses provide assurance that our GPS effect estimates among QV carriers are not biased by our original risk score design.

### CKD phenotyping and case-control definitions

We used our validated CKD e-phenotyping algorithm to define CKD cases and controls[33]. All cases had either estimated glomerular filtration rate (eGFR) below 60 ml/min/1.73 m² (by 2009 CKD-EPI equation[34]) or received a renal replacement therapy (dialysis or kidney transplant). All controls had eGFR greater than 90 ml/min/1.73 m² and no evidence of CKD based on diagnostic or procedure billing codes. Similar to our prior studies, we excluded individuals with eGFR 60–90 ml/min/1.73 m² from case-control cohorts in order to minimize potential case-control misclassification due to age-related decline in eGFR[30]. The covariates included age, sex, diabetes (type I or type II) defined based on ICD codes[6], and significant principal components of ancestry, similar to our prior validation studies[30].

### Predictive performance

The predictive performance of the GPS was assessed using standardized metrics as recently proposed by ClinGen[59], including area under the receiver operating characteristics curve (AUROC), variance explained ($R^2$), and effect size (OR) per standard deviation of the GPS distribution in controls. We used the pROC R package to calculate AUROC. For effect size estimation, we used logistic regression (glm function in R) with CKD status as an outcome and standardized GPS as a predictor with adjustment for age, sex, diabetes mellitus (type I or type II), genotype/imputation batch, and four PCs of ancestry, similar to prior studies[30]. Similarly, the association of a carrier status with CKD was tested using a logistic regression with CKD case status as an outcome and carrier status as a predictor, controlling for age, sex, diabetes, batch, and ancestry PCs. The same logistic model with the included GPS and carrier status terms was used to test the GPS-by-carrier status interaction. To compare GPS effect sizes between carriers and noncarriers, we derived ORs (and 95% CIs) of CKD, comparing each tertile of the GPS distribution in the carriers to the reference middle (2nd) tertile of the GPS for noncarriers in each cohort. For all analyses, we used R version 4.2.2 (2022-10-31).

### Meta-PheWAS

We performed a phenome-wide association analysis for ADPKD and COL4A-AN variant carriers in both AoU and UKBB datasets. The 165,208 genotyped and imputed AoU participants had 12,945 ICD-9 codes that were first mapped to 1817 distinct phecodes. Similarly, there were 10,221 ICD-9 codes for UK Biobank participants ($N = 460,363$) with imputed genotype data that mapped to 1817 distinct phecodes. Phenome-wide associations were performed using the PheWAS R package[60]. The package uses two ICD-9 codes occurrences within a given phecode grouping to define a case and pre-defined "control" groups for each phecode. All 1817 phecodes were tested using logistic regression with case-control status as the outcome and genotype, sex, age, batch, and five principal components of ancestry as predictors. We then performed fixed-effects Meta-PheWAS of AoU and UKBB datasets using the PheWAS R package. We set the Bonferroni corrected statistical significance threshold for phenome-wide significance at 2.75E-05 (0.05/1817 phecodes tested).

### Reporting summary

Further information on research design is available in the Nature Portfolio Reporting Summary linked to this article.

## Data availability

The UKBB genotype and phenotype data are available through the UKBB web portal at https://www.ukbiobank.ac.uk/. All researchers who wish to access the research resource must register with the UK Biobank. The AoU genotype, WGS, and phenotype data are available through the AoU researcher workbench at https://www.researchallofus.org/data-tools/workbench/. The researchers interested in accessing these data must complete registration with the AoU study. Both biobanks require institutional data use agreements as part of the registration process. The variants included in the analyses under various models are provided in Supplementary Data 1 and 2; meta-PheWAS summary statistics are provided in Supplementary Data 3 and 4. Any additional results and data supporting the findings described in this manuscript are available in the article and its Supplementary Information files and from the corresponding author upon request.

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

## Acknowledgements

This work was funded by the National Human Genome Research Institute (NHGRI) Electronic Medical Records and Genomics-IV (eMERGE-IV grant 5U01HG008680-07), National Library of Medicine (NLM grant R01LM013061), National Institute of Diabetes, Digestive and Kidney Diseases (NIDDK grant 5K25DK128563-03) and National Center for Advancing Translational Sciences (NCATS grant UL1TR001873). We are grateful to all participants from the UKBB and AoU projects for contributing their data and biological samples that enabled this study. The research on UK Biobank data has been conducted using the UK Biobank Resource under Application Number 41849. The All of Us Research Program is supported by the NIH Office of the Director through the following grants: Regional Medical Centers: 1OT2OD026549; 1OT2OD026554; 1OT2OD026557; 1OT2OD026556; 1OT2OD026550; 1OT2OD 026552; 1OT2OD026553; 1OT2OD026548; 1OT2OD026551; 1OT2OD026555; IAA# AOD16037; Federally Qualified Health Centers: HHSN 263201600085U; Data and Research Center: 5U2COD023196; Biobank: 1U24OD023121; The Participant Center: U24OD023176; Participant Technology Systems Center: 1U24OD023163; Communications and Engagement: 3OT2OD023205; 3OT2OD023206; and Community Partners: 1OT2OD025277; 3OT2OD025315; 1OT2OD025337; 1OT2OD025276.

## Author contributions

Project conceptualization: A.K., K.K.; analytical methods and genetic data analysis: A.K., K.K.; electronic phenotyping: A.K., N.S., C.W., G.H., J.G.N., monogenic variant filtering methods and review: A.G., P.C.H.; manuscript draft: A.K., K.K.; overall project supervision: K.K.

## Competing interests

The authors declare no competing interests.
