## [Peer Review File · Nature Communications]

Polygenic risk alters the penetrance of monogenic kidney diseaseREVIEWER COMMENTS

Reviewer #1 (Remarks to the Author):

This paper examines the role of a CKD polygenic risk score (PRS) previously developed by this group in risk of monogenic disease, specifically polycystic kidney disease (PKD1, PKD2) and COL4-associated nephropathy (Alport syndrome).

This is an interesting, well-executed, and important paper. The authors find a remarkably strong effect of the PRS on penetrance of PKD alleles, and a strong, though smaller, effect on COL4 alleles. It is interesting that the same PRS score seems to modify the risk associated with these two quite different monogenic disease. Since this PRS has effects on risk of CKD in people with PKD alleles, COL4 alleles, and without monogenic risk alleles, these PRS presumably reflect some polygenic propensity towards kidney function decline, rather than a genetic propensity for something like kidney cyst growth. It also raises the possibility that an even better PKD-specific PRS could be developed.

I think this paper will be of interest to both the nephrology and genetics communities.

I think this paper is methodologically sound and that the authors provide sufficient detail for others to reproduce and/or apply these findings.

Reviewer #2 (Remarks to the Author):

This dive into CKD genetic architecture is a nice step to better understanding complex traits, leveraging CKD's mix of big and small effects. The paper's strength is using straightforward and valid tests of straightforward and well-motivated hypotheses. My only major concern is about genetic and non genetic heterogeneity across populations. I also suggested some further statistical tests that could strengthen the paper. Overall, I enjoyed and learned from reading this paper.

Major: The PGS is not portable. I couldn't tell from this paper, but it is clear from Khan NatMed paper (Fig 3b). I **really** appreciate and support the effort mitigate PGS bias (which is due to biases in the data

itself). But that doesn't imply that the PGS is portable. To be clear, the more-portable PGS is a strength of this paper, but all the standard concerns about PGS portability must still be addressed.

First, I think this needs to be more clear in the writing throughout, eg the abstract says the PGS is 'independent of age, sex, diabetes, and genetic ancestry' which is misleading (the result itself is great, but this phrasing implies the GPS works equally for everyone, which is not true). There are several other examples.

Second, robustness tests are needed (which will also add robustness for the rare risk variants, which have vaguely similar biases as the PGS).

-The simplest secondary analysis idea is to just rerun analyses after subsetting to the largest populations (EUR/whiteBritish). This will dramatically reduce potential for biases without adding new analyses or sacrificing too much power.

-Even better is to analyze all populations (as defined by the 1KG clustering, which is a nice data-driven/agnostic way to align across datasets). I don't know if this will have power--but even if they are underpowered, it is still good to report this consequence of our biased underlying data.

-Even better is to evaluate interactions between PGS/rare variants and population labels (eg with meta-analysis), but this is likely out of scope. If you do go forward with this, be careful with interpretation, as interactions are MUCH likelier due to nongenetic population differences rather than genetic ones (especially with eGFR).

Overall, I support the goal of finding a single PGS that can be used for everyone in the clinic, but in basic science analyses I think it's important to account for genetic and non genetic differences between populations to avoid confounding biases, especially given eGFR phenotyping biases and PGS biases.

Related major point: eGFR phenotyping. Are you using the 'race-adjusted' eGFR phenotypes? These seem to be the main focus of the NatMed paper, though that paper also discusses and compares to other eGFR measures. But I don't see any of this in this paper. This seems very important for understanding the portability of your results and also the estimated effect sizes of any variables that differ as a function of this 'race' variable, eg GPS distribution or effect size (the NatMed paper says the former is not very different but the latter is very different), or rare variant frequencies/effect sizes (it's clear the former vary between populations). The choices made with this phenotype definitely have to be explained and justified clearly, both statistically and ethically. Even better, it seems like the same robustness test as in SuppTab10 of the NatMed paper is even more warranted than in the original paper (given the complexities of rare variants). It is not obvious to me that the rare variants will perform similarly to the GPS.

Also, I think it's essential to comment on the idea of race correction in eGFR in the Discussion. In particular, I think it must be clear to readers that this article is not really impacting that debate one way or another--rather, it takes the phenotype as-is and assumes that debate is irrelevant to the genetic results in this study ('assumes' can be upgraded to 'likely' if you perform the suggested sensitivity analysis akin to SuppTab10 in NM paper).

Minor comments

-Why is Apol1 not considered alongside the other large-effect genes, but rather in the GPS? This is partly done in the NM paper (STab2)--but the allosus data will add substantial power. Related, why not include the large effect genes in your GPS and test the performance improvement? This seems to be (one of) the main point of the NM paper. (Related, it's far from clear to me that transforming the GPS to a standard normal in each population is the right thing to do, especially if Apol1 is population specific and has a big effect...but I think that dealing with this is out of scope for the current paper.)

-It's ok that the GPS-by-carrier interaction tests are null, but please report all of them. Related, please report how you encoded 'carrier', and ideally return these to UKB (I would use it). Finally, please evaluate the power of these tests given the carriers are rare and high-risk.

-PheWAS is a great way to validate the EHR phenotypes. But this could be taken further to answer more novel questions. For example, I think these are pretty straightforward: stratify by carrier status; stratify by risk gene; stratify by population; run PheWAS of carrier status scores.

-"the GPS significantly alters the penetrance of ADPKD M1 qualifying variants"--I don't think this is true, the interaction isn't significant. It does modify risk the these carriers, but that's different than modifying the penetrance. This concept pops up several times and ~half of the time has this issue. An example of correct phrasing is 'we test if this GPS modifies the risk of CKD among carriers', another incorrect example is "This effect was most pronounced in the individuals with known pathogenic or rare pLOF variants in PKD1 or PKD2".

-I believe the M2 contains M1 and M3 contains M1+M2. It would be interesting to run each set separately as a secondary analysis to understand their relative effect sizes--especially in the analysis where M1 and M3 are significant but M2 is not, what is going on there?

-It would be nice to see numbers on the scale of absolute risk since the effect sizes are so large. I.e. X% of M1 carriers have CKD vs X% of the middle GPS tertile. This is not necessary but will really help people understand the take home messages. It also is a way of correcting statements like "This effect was most pronounced in the individuals with known pathogenic or rare pLOF variants in PKD1 or PKD2"--on the absolute scale, this is likely true.

-Did you remove related individuals in UKB? I assume so (this is necessary, and is done in allofus) but didn't see it described.

-How were MAF filters applied? "filtered variants based on frequency in any ancestral groups" seems unlikely, as the *apol1* variants have ~0% prevalence in European populations. I think the requirement should be (and probably is) that the MAF is sufficiently high in at least one group.

-Why use a 'biallelic (recessive)' encoding for *col4a*? I don't see any rationale either for the biallelic or the recessive, nor any exploration of sensitivity. Generally, this paper misses opportunities to test relationships between rare variants genes. (But there are solid efforts to test sensitivity to the definition of risk variants themselves, and the subgroup analyses across genes).

-'trans-ancestry' -- please see <https://www.nature.com/articles/s41588-021-00952-6>

-"Because the risk of CKD increases with age, these differences may be partially responsible for a lower effect estimate for the GPS in the AoU compared to the UKBB dataset." I would bet a lot of money this is because your GPS is not portable between ancestries (UKBB is vast majority Eur-anc). It is misleading to reference these features, which you have not proven modify GPS portability, but to ignore the portability bias, which is unambiguous. There are also many differences between the US and UK. Finally, it is not necessarily true that age will change GPS portability--this only happens if there is an interaction, a mere additive effect of age is insufficient (so long as appropriate metrics are used).

-Why are the sample sizes so low? Why can't you calculate eGFR on everyone? Related, the Introduction sentence " a total of 568,457 participants" is misleading as it is roughly 2x compared to the Results statement: "we defined a total of 10,081 CKD cases and 266,724 controls in the UKBB and 11,820 CKD cases and 22,763 controls"

Other

-I don't love the plotting style in Figs 3b/4b. Consider instead plotting the normal odds ratios--currently, I believe all are with respect to the non-carrier middle-tertile, but the more natural comparison is carrier at tertile X vs non carrier at tertile X. I think this will make it easier to think about the interaction effect--but I acknowledge it will make it harder to compare to the 'population average', so I understand if you disagree. Ideally you make one version a main figure and the other a supplementary figure.

-In the future, I think you could likely improve your GPS using methods to optimize portability, eg <https://www.nature.com/articles/s41467-023-36544-7> or <https://doi.org/10.1101/2023.04.12.536510>

Reviewer #3 (Remarks to the Author):

This study aims to examine whether an individual's risk of developing kidney disease is influenced by their polygenic genetic background for individuals with rare pathogenic variants in monogenic disease-causing genes. The study utilized genomic and phenotypic data from the UK Biobank (UKB) and All of Us (AoU) datasets and conducted a corresponding meta-analysis. The researchers used the GPS model they had constructed for CKD and categorized rare variants of ADPKD and COL4A-AN genes into three models to test the additive effects of PRS with rare pathogenic variants, also with sensitivity analysis. The data analysis is rigorous and solid, and the manuscript is well-written. It's joyful to review this manuscript, just with a few minor comments.

1. Since including the overlapped samples used for deriving GWAS summary and the samples for studying GPS performance will lead to over-estimate the PGS effect size, so would suggest stating clearly in that text that there's no sample overlapping of these samples.

2. Since traditional logistic regression models were applied, using unrelated samples in the data analysis should be better. So more than removing the related samples by the KING coefficient of 0.35 is required. It's better to clean to the 2nd or even 3rd degree of relationships.

3. In the RESULTS: "we defined a total of 10,081 CKD cases and 266,724 controls in the UKBB and 11,820 CKD cases and 22,763 controls in the AoU", the number is much smaller than the number in Fig 1. If this

is a QC-related problem, suggest listing the QC procedures and samples removed in each step in the Methods part or in the flowchart figure.

4. In Table 1, since it's a meta-analysis, suggests listing the effect size and carrier numbers estimated from each data set.

Reviewer #1

This paper examines the role of a CKD polygenic risk score (PRS) previously developed by this group in risk of monogenic disease, specifically polycystic kidney disease (PKD1, PKD2) and COL4-associated nephropathy (Alport syndrome).

This is an interesting, well-executed, and important paper. The authors find a remarkably strong effect of the PRS on penetrance of PKD alleles, and a strong, though smaller, effect on COL4 alleles. It is interesting that the same PRS score seems to modify the risk associated with these two quite different monogenic diseases. Since this PRS has effects on risk of CKD in people with PKD alleles, COL4 alleles, and without monogenic risk alleles, these PRS presumably reflect some polygenic propensity towards kidney function decline, rather than a genetic propensity for something like kidney cyst growth. It also raises the possibility that an even better PKD-specific PRS could be developed.

I think this paper will be of interest to both the nephrology and genetics communities. I think this paper is methodologically sound and that the authors provide sufficient detail for others to reproduce and/or apply these findings.

We appreciate the reviewer's thoughtful remarks and kind comments.

Reviewer #2

This dive into CKD genetic architecture is a nice step to better understanding complex traits, leveraging CKD's mix of big and small effects. The paper's strength is using straightforward and valid tests of straightforward and well-motivated hypotheses. My only major concern is about genetic and non-genetic heterogeneity across populations. I also suggested some further statistical tests that could strengthen the paper. Overall, I enjoyed and learned from reading this paper.

Thank you for sharing your valuable expertise and constructive feedback.

Major:

The PGS is not portable. I couldn't tell from this paper, but it is clear from Khan NatMed paper (Fig 3b). I *really* appreciate and support the effort mitigate PGS bias (which is due to biases in the data itself). But that doesn't imply that the PGS is portable. To be clear, the more-portable PGS is a strength of this paper, but all the standard concerns about PGS portability must still be addressed.

First, I think this needs to be more clear in the writing throughout, eg the abstract says the PGS is 'independent of age, sex, diabetes, and genetic ancestry' which is misleading (the result itself is great, but this phrasing implies the GPS works equally for everyone, which is not true). There are several other examples.

We greatly appreciate this feedback, and we agree with these comments. In response, we have carefully revised the abstract and the entire manuscript to address these concerns, especially in the way we discuss the portability issue. Please note that the abstract was also shortened to under 200 words as per the journal's guideline.

Second, robustness tests are needed (which will also add robustness for the rare risk variants, which have vaguely similar biases as the PGS).

-The simplest secondary analysis idea is to just rerun analyses after subsetting to the largest populations (EUR/whiteBritish). This will dramatically reduce potential for biases without adding new analyses or sacrificing too much power.

-Even better is to analyze all populations (as defined by the 1KG clustering, which is a nice data-driven/agnostic way to align across datasets). I don't know if this will have power--but even if they are underpowered, it is still good to report this consequence of our biased underlying data.

-Even better is to evaluate interactions between PGS/rare variants and population labels (eg with meta-analysis), but this is likely out of scope. If you do go forward with this, be careful with interpretation, as interactions are MUCH likelier due to nongenetic population differences rather than genetic ones (especially with eGFR).

In response, we conducted additional sensitivity analysis for the most powerful European ancestry subset of the UKBB. We observed consistent results with our overall meta-analyses, providing reassurance that our results are not biased by population stratification or ancestry adjustments. These results, including the breakdown by ADPKD and COL4A-AN, are now summarized in the new Supplementary Table 10. Unfortunately, as the reviewer points out, we are presently underpowered to perform similar analyses in non-European population clusters.

Related major point: eGFR phenotyping. Are you using the 'race-adjusted' eGFR phenotypes? These seem to be the main focus of the NatMed paper, though that paper also discusses and compares to other eGFR measures. But I don't see any of this in this paper. This seems very important for understanding the portability of your results and also the estimated effect sizes of any variables that differ as a function of this 'race' variable, eg GPS distribution or effect size (the NatMed paper says the former is not very different but the latter is very different), or rare variant frequencies/effect sizes (it's clear the former vary between populations). The choices made with this phenotype definitely have to be explained and justified clearly, both statistically and ethically. Even better, it seems like the same robustness test as in SuppTab10 of the NatMed paper is even more warranted than in the original paper (given the complexities of rare variants). It is not obvious to me that the rare variants will perform similarly to the GPS. Also, I think it's essential to comment on the idea of race correction in eGFR in the Discussion. In particular, I think it must be clear to readers that this article is not really impacting

that debate one way or another--rather, it takes the phenotype as-is and assumes that debate is irrelevant to the genetic results in this study ('assumes' can be upgraded to 'likely' if you perform the suggested sensitivity analysis akin to SuppTab10 in NM paper).

In our primary analyses, we used the 2009 CKD-EIP eGFR equation for consistency with our Nature Medicine publication and because we had previously demonstrated that the equation itself had no major effect on the performance of the GPS. However, at the reviewer's request, we have now added new sensitivity analyses using the updated (2021) CKD-EPI equation. As expected, fewer CKD cases were identified using the new equation, especially in the analysis of the European subgroup (Supplemental Tables 9 and 10). Despite smaller case counts, however, our risk estimates were consistent across the GPS tertiles for both ADPKD and COL4A-AN. These analyses demonstrate that our results are robust to the equation choice.

Minor:

-Why is Apol1 not considered alongside the other large-effect genes, but rather in the GPS? This is partly done in the NM paper (STab2)--but the alofus data will add substantial power. Related, why not include the large effect genes in your GPS and test the performance improvement? This seems to be (one of) the main point of the NM paper. (Related, it's far from clear to me that transforming the GPS to a standard normal in each population is the right thing to do, especially if Apol1 is population specific and has a big effect...but I think that dealing with this is out of scope for the current paper.)
-It's ok that the GPS-by-carrier interaction tests are null, but please report all of them. Related, please report how you encoded 'carrier', and ideally return these to UKB (I would use it). Finally, please evaluate the power of these tests, given the carriers are rare and high-risk.

We appreciate the suggestion, but the magnitude of effect for the APOL1 risk genotype alone is considerably smaller in comparison with the monogenic variants (~50% increased risk for the APOL1 high-risk genotype compared to >3-fold increased risk for monogenic variants); thus, we decided to only use the combined GPS approach as initially proposed in our Nature Medicine paper. We also decided not to incorporate monogenic variants within the GPS, given that we do not have external cohorts to further re-optimize the weights, or properly test the performance of a combined score. This could be potentially done with the use of future releases of the AoU data, but we feel is out of scope for the current paper. Regarding your second suggestion, we have now included the effect estimates and P-values for the GPS-by-carrier interactions for both ADPKD (Supplemental Table 7) and COL4A-AN (Supplemental Table 12). We added a clear statement to the discussion that we are underpowered for these tests given the limited number of carriers identified in our datasets.

-PheWAS is a great way to validate the EHR phenotypes. But this could be taken further to answer more novel questions. For example, I think these are pretty straightforward: stratify by carrier status; stratify by risk gene; stratify by population; run PheWAS of carrier status scores.

Thank you for these suggestions. Indeed, we have performed most of the suggested sensitivity analyses to internally check for biases when preparing the manuscript, but these were not included in our first submission. We have now included these analyses in the revised manuscript in Supplementary Figures 7-8.

-"the GPS significantly alters the penetrance of ADPKD M1 qualifying variants"--I don't think this is true, the interaction isn't significant. It does modify risk the these carriers, but that's different than modifying the penetrance. This concept pops up several times and ~half of the time has this issue. An example of correct phrasing is 'we test if this GPS modifies the risk of CKD among carriers', another incorrect example is "This effect was most pronounced in the individuals with known pathogenic or rare pLOF variants in PKD1 or PKD2".

We agree that careful phrasing of these results is important, and we have now corrected some of these statements, and we added our definition of penetrance to the methods section. We defined penetrance as the probability of CKD conditioned on the QV carrier status. We observed that the risk of CKD varied as a function of polygenic risk among the QV carriers. Thus, the conclusion that the GPS alters the penetrance of QV variants is in our view justified. We specifically avoided the term "modifies the penetrance" since this would imply a non-additive (interaction) effect (i.e., effect modification), whereas the observed change was purely additive. To reduce any confusion, we have now provided additional corrections to the phrasing and clarifications in the manuscript.

-I believe the M2 contains M1 and M3 contains M1+M2. It would be interesting to run each set separately as a secondary analysis to understand their relative effect sizes--especially in the analysis where M1 and M3 are significant but M2 is not, what is going on there?

Thank you for this question, which allows us to clarify this point. While it is true that M2 contains M1 variants, it is important to note that M3 does not necessarily include all variants from M1 and M2, which explains why M2 and M3 results could differ. This distinction arises from our utilization of a distinct approach to define M3. Specifically, we exclusively incorporate variants designated as deleterious by all five prediction algorithms and a REVEL score ≥ 0.7 , but did not automatically include all ClinVar 'P' and 'LP' variants. We have added the following clarification under our model definitions in the methods section: "Notably, while M2 contains all M1 variants, the definition of M3 does not necessarily encompass all M1 and M2 variants."

-It would be nice to see numbers on the scale of absolute risk since the effect sizes are so large. I.e X% of M1 carriers have CKD vs X% of the middle GPS tertile. This is not necessary but will really help people understand the take home messages. It also is a way of correcting statements like "This effect was most pronounced in the individuals with known pathogenic or rare pLOF variants in PKD1 or

PKD2"--on the absolute scale, this is likely true.

Thank you for this suggestion. We have now incorporated this information into the manuscript.

-Did you remove related individuals in UKB? I assume so (this is necessary, and is done in all of us) but didn't see it described.

Indeed, we had addressed this issue in our QC analyses, but this was not described, so thanks for pointing out this omission. We have now added this information to the methods section.

-How were MAF filters applied? "filtered variants based on frequency in any ancestral groups" seems unlikely, as the *apol1* variants have ~0% prevalence in European populations. I think the requirement should be (and probably is) that the MAF is sufficiently high in at least one group.

Sorry for the confusion. The goal was to retain only very rare, ultrarare, or novel variants that were not exceeding a given MAF threshold in any individual ancestral group. This is a very stringent filtering approach. *APOL1* risk variants would be eliminated by these filters because their MAF exceeds our thresholds in African ancestry subgroups. We have now re-phrased this statement in the methods to avoid any confusion.

-Why use a 'biallelic (recessive)' encoding for *col4a*? I don't see any rationale either for the biallelic or the recessive, nor any exploration of sensitivity. Generally, this paper misses opportunities to test relationships between rare variants genes. (But there are solid efforts to test sensitivity to the definition of risk variants themselves, and the subgroup analyses across genes).

The rationale for examining recessive encoding for *COL4A* is that recessive inheritance underlies Alport syndrome, the most severe form of *COL4A*-AN that is characterized by a more rapid progression of CKD. We perform this analysis only on an exploratory basis because, as expected, we observe only a very small number of such cases. The limited sample size currently precludes a more detailed analysis of individual genes under this model.

-'trans-ancestry' -- please see <https://www.nature.com/articles/s41588-021-00952-6>

Thanks for this comment, we removed the term trans-ancestry from the manuscript.

-"Because the risk of CKD increases with age, these differences may be partially responsible for a lower effect estimate for the GPS in the AoU compared to the UKBB dataset." I would bet a lot of money this is because your GPS is not portable between ancestries (UKBB is vast majority Eur-anc). It is misleading to reference these features, which you have not proven modify GPS portability, but to ignore the portability bias, which is unambiguous. There are also many differences between the US and UK. Finally, it is not necessarily true that age will change GPS portability--this only happens if there is an interaction, a mere additive effect of age is insufficient (so long as appropriate metrics are used).

Thanks for this comment; we completely agree that the ancestry portability issue is likely contributing. We have now changed the way we discuss the observed differences between the biobanks.

-Why are the sample sizes so low? Why can't you calculate eGFR on everyone? Related, the Introduction sentence "a total of 568,457 participants" is misleading as it is roughly 2x compared to the Results statement: "we defined a total of 10,081 CKD cases and 266,724 controls in the UKBB and 11,820 CKD cases and 22,763 controls"

We apologize if these numbers were misleading, but we processed all available genotype and sequence data for the UKBB and All-of-Us cohorts before linking these datasets to the available phenotypes, thus we need to refer to the total number of participants when discussing QC of the genetic data. However, in the final analysis we include only the subset of individuals with available phenotype data that meets our strict inclusion/exclusion criteria. The reduced numbers of individuals in the final analysis are predominantly due to missing lab data (required for eGFR calculation). For instance, approximately 50% of the release 1 All-of-Us participants currently lack lab data including serum Cr values for eGFR calculations (we were also surprised by this!). Furthermore, similar to our prior studies, we excluded any individuals with eGFR in the range 60-90 mL/min/1.73m² to minimize potential case-control misclassification. We also excluded any individuals that could not be classified as cases or controls with certainty (e.g., individuals with eGFR > 90 mL/min/1.73m² but a single ICD code suggestive of any form of kidney disease would be excluded from the control group). We also exclude all cryptically related individuals, and those that do not have both SNP array AND sequence data after the QC. We have now provided clarifications to justify the final case-control numbers in the methods section of the manuscript.

Other:

-I don't love the plotting style in Figs 3b/4b. Consider instead plotting the normal odds ratios--currently, I believe all are with respect to the non-carrier middle-tertile, but the more natural comparison is carrier at tertile X vs non carrier at tertile X. I think this will make it easier to think about the interaction effect--but I acknowledge it will make it harder to compare to the 'population average', so I understand if you disagree. Ideally you make one version a main figure and the other a supplementary figure.

We agree that the visual presentation of these results is challenging, but after exploring various styles of data visualization, we have chosen the current format that we feel illustrates our results most effectively. We agree that comparing carriers to non-carriers within the same tertile would better visualize potential interactions, but these were not significant. Also, this visualization would not illustrate the gradient of risk by GPS among the carriers in relationship to the 'average risk'. Therefore, we prefer to keep our original presentation.

-In the future, I think you could likely improve your GPS using methods to optimize portability, eg <https://www.nature.com/articles/s41467-023-36544-7> or <https://doi.org/10.1101/2023.04.12.536510>

Thank you for this suggestion, and we completely agree. As you are well aware, this is an area of active research, and we have every intention to incorporate these and other methods to further enhance the portability of our GPS in the near future. For now, we have added these references to the discussion.

Reviewer #3

This study aims to examine whether an individual's risk of developing kidney disease is influenced by their polygenic genetic background for individuals with rare pathogenic variants in monogenic disease-causing genes. The study utilized genomic and phenotypic data from the UK Biobank (UKB) and All of Us (AoU) datasets and conducted a corresponding meta-analysis. The researchers used the GPS model they had constructed for CKD and categorized rare variants of ADPKD and COL4A-AN genes into three models to test the additive effects of PRS with rare pathogenic variants, also with sensitivity analysis. The data analysis is rigorous and solid, and the manuscript is well-written. It's joyful to review this manuscript, just with a few minor comments.

Thank you for your kind feedback.

1. Since including the overlapped samples used for deriving GWAS summary and the samples for studying GPS performance will lead to over-estimate the PGS effect size, so would suggest stating clearly in that text that there's no sample overlapping of these samples.

Thank you for highlighting this aspect. We agree with your observation and have incorporated a corresponding statement in the manuscript: "It is also important to note that the summary statistics used for the development of the GPS were derived from completely separate GWAS cohorts than the ones used for testing the GPS performance."

2. Since traditional logistic regression models were applied, using unrelated samples in the data analysis should be better. So more than removing the related samples by the KING coefficient of 0.35 is required. It's better to clean to the 2nd or even 3rd degree of relationships.

Thank you for this comment; we have now taken steps to clarify this in the revised manuscript. Please see our responses to reviewer 2 for more details.

3. In the RESULTS: "we defined a total of 10,081 CKD cases and 266,724 controls in the UKBB and 11,820 CKD cases and 22,763 controls in the AoU", the number is much smaller than the number in Fig 1. If this is a QC-related problem, suggest listing the QC procedures and samples removed in each step in the Methods part or in the flowchart figure.

Thank you for bringing this to our attention. The requested clarification has been included in the manuscript. Please see our response to Reviewer 2.

4. In Table 1, since it's a meta-analysis, suggests listing the effect size and carrier numbers estimated from each data set.

Thank you for the suggestion. We have incorporated the effect sizes and carrier numbers for each dataset (UKBB and AoU followed by meta-analysis) into the revised Table 1.

REVIEWERS' COMMENTS

Reviewer #2 (Remarks to the Author):

I think this is an excellent revision that addresses all of my concerns. The added sensitivity analyses reassure me that the results are not substantially biased by population stratification. I also appreciate the small writing additions to reinforce that PGS portability and equitable CKD diagnosis remain important open problems. (I recognize that all of this was likely already obvious to the authors, but I appreciate their effort to clarify to readers.)

Reviewer #3 (Remarks to the Author):

No further comments